# Recent Attempts in the Design of Efficient PVC Plasticizers with Reduced Migration

**DOI:** 10.3390/ma14040844

**Published:** 2021-02-10

**Authors:** Joanna Czogała, Ewa Pankalla, Roman Turczyn

**Affiliations:** 1Department of Physical Chemistry and Technology of Polymers, Silesian University of Technology, Strzody 9, 44-100 Gliwice, Poland; 2Research and Innovation Department, Grupa Azoty Zakłady Azotowe Kędzierzyn S.A., Mostowa 30A, 47-220 Kędzierzyn-Koźle, Poland; Ewa.Pankalla@grupaazoty.com; 3Joint Doctoral School, Silesian University of Technology, Akademicka 2A, 44-100 Gliwice, Poland

**Keywords:** PVC, primary plasticizer, migration resistance, bio-based plasticizers, plasticizing efficiency

## Abstract

This paper reviews the current trends in replacing commonly used plasticizers in poly(vinyl chloride), PVC, formulations by new compounds with reduced migration, leading to the enhancement in mechanical properties and better plasticizing efficiency. Novel plasticizers have been divided into three groups depending on the replacement strategy, i.e., total replacement, partial replacement, and internal plasticizers. Chemical and physical properties of PVC formulations containing a wide range of plasticizers have been compared, allowing observance of the improvements in polymer performance in comparison to PVC plasticized with conventionally applied bis(2-ethylhexyl) phthalate, di-n-octyl phthalate, bis(2-ethylhexyl) terephthalate and di-n-octyl terephthalate. Among a variety of newly developed plasticizers, we have indicated those presenting excellent migration resistance and advantageous mechanical properties, as well as those derived from natural sources. A separate chapter has been dedicated to the description of a synergistic effect of a mixture of two plasticizers, primary and secondary, that benefits in migration suppression when secondary plasticizer is added to PVC blend.

## 1. Introduction

Polymers are a large group of materials that are used in everyday life. To increase their functionality, they are usually modified with additives to induce desired properties, i.e., thermal stability, softness, electrical insulating, flame retardancy, etc. According to plasticseurope.org, in 2019 the global plastic production was equal to 368 million tonnes, and it was 9 million tonnes higher than in 2018. Ten percent of this number was assigned to poly(vinyl chloride) (PVC), which makes it the fourth most common plastic in the world. PVC is usually applied in the production of cables, floorings, packaging, medical devices, toys, etc. [1,2]

Plasticizers are the most popular plastic additive that makes materials more flexible and easier to process, and about 90% of them are used for PVC applications. Traditionally, the most common PVC plasticizers are phthalates. Unfortunately, in the last few years, serious environmental protection and society health hazard issues have been raised, mainly because phthalate plasticizers have been found in soils, sea water and sediments, in leaving organisms and even in human breast milk [3,4,5,6,7]. Under increasing environmental awareness and social pressures, phthalate plasticizers have become a subject of legal restrictions in many countries. Even though traditional phthalate plasticizers are still the most frequently used plasticizers around the world, there is an urgent need to replace them with new, less harmful materials. Therefore, researchers have nowadays chosen to focus on the development of novel plasticizers and/or plasticizing systems without adverse effects on the environment and, at the same time, without a deteriorating influence on the properties of the final product [8,9].

### 1.1. Plasticization Mechanism

A plasticization theory was developed in the 40’s and 50’s of the last century and the mechanism of plasticization has been explained in several approaches:Gel theory [10,11,12] stating that a plasticizer reduces interaction between chains of a polymer three-dimensional network.Lubricity theory [13] stating that plasticizer molecules act as a lubricant between polymer chains and enable them to slip.Free volume theory [14,15], stating that plasticizers increase the free space in a polymer matrix giving the polymer chains more space to move.

The last one, proposed by Fox and Flory in 1950’s, is the most known and useful theory, giving a suitable explanation for the plasticization phenomenon. The free volume can be understood as a gap between molecules and atoms in a temperature above the glass transition temperature with subtraction of the solid specific volume extrapolated to a particular temperature. The addition of a plasticizer increases the free volume, which implies increasing motion of polymer chains. During the years the theory has evolved. Sears and Darby [16] presumed that the free volume comes from three main sources: the motion of chain ends, the motion of side chains and the motion of the main chain. The movement of the polymer also depends on the temperature and increases with increasing temperature making plastics more flexible [17]. It is worth mentioning that usually plasticizers with a small molecular size add a greater free volume to the system than large compounds. Moreover, a branched plasticizer is more effective in increasing the free volume than a linear compound. Therefore, to be more efficient in providing free volume, a plasticizer should have a reasonably low molecular weight but a relatively big molecular size [8,18,19,20].

Because of the inter-chain interactions of polar carbon-chlorine bonds in PVC, the resin without a plasticizer is brittle, hard and causes difficulties during processing. The aim of the use of plasticizers is to enhance the flexibility and processability of polymers, what is achieved by lowering the second order glass transition temperature of material. The functional properties of plasticized PVC plastics are closely related to the PVC backbone–plasticizer molecule interactions, as well as intermolecular PVC chain-to-chain coupling. There exist numerous structures of plasticizer molecules that provide beneficial interactions with the PVC chain, like ester groups, epoxy groups, aromatic rings, hydroxyl groups, etc. A relationship between chemical structure of a plasticizer and its “goodness” was discussed in 2016 by M. Bocqué et al. [8] Three main building blocks of a plasticizer molecule were distinguished: (1) spacer—usually aliphatic chains that favour intermolecular non-polar interactions; (2) cohesive block—e.g., ester groups; and (3) compatibilizer blocks—an aromatic ring. The nature of these structure elements, as well as the molecular weight of molecules, implies plasticizer efficiency and its migration resistance. The most common types of intermolecular forces of attraction present in PVC–plasticizer blends are van der Waals forces, including dipole–dipole and dipole-induced dipole attractions, London dispersion forces, as well as hydrogen bonds. The dipole–dipole interactions occur between the polar groups of plasticizers, like carbonyl in ester linkages [21] or oxygen in Si-O-Si linkages [22] and chlorine atoms in PVC backbone. In a plasticizer molecule, a hydrogen bonding may appear between the carbonyl groups of the ester and the hydrogen attached to the carbon in an α position relative to the carbon–chlorine bonds in PVC [21]. These interactions reduce the original dipole-induced dipole attractions among polymer chains and reduce the entanglement of polymer chains, thus changing the 3D molecular organization of polymers by increasing its mobility [23].

A convenient technique to investigate the interactions and miscibility between polymer and plasticizer is FTIR spectroscopy. An efficient plasticizer interacts with PVC and breaks the extended network of chain-to-chain dipole-induced dipole forces present originally in a polymer that is observed as an attenuation of a signal coming from the covalent bonds. The stronger the interactions, the more prominent the shift of particular peaks to the lower wavenumber [24]. Signals of a particular interest are methylene groups and carbon-chlorine bonds stretching vibrations at 1450–1400 cm^−1^ and 650–600 cm^−1^, respectively, which are originated from the crystalline domains of PVC. Plasticization of PVC results in a weakening of the intensity of both these bands [25]. Moreover, the changes in spectral features of the plasticizer molecule can be followed to evaluate its interactions with PVC. For instance, for an ester type plasticizers, signals connected to the C-O-C and carbonyl group stretching vibrations shift to the lower wavenumber because of an increase in an electron density of ester group and dipole–dipole interaction between hydrogen atoms at the α position of PVC and plasticizer’s carbonyl group [20,22,26].

### 1.2. Chemical Classes of Plasticizers

There are a few chemical structures that provide good plasticizing efficiency. Comparing different types of plasticizers, a relationship between the presence of functional groups, chain branching and their influence on plastics properties can be investigated. Dangling chains, especially those aliphatic in nature, act as spacers able to induce high motion dynamics, providing good plasticizing efficiency. Being a nonpolar (apolar) structural component, they add the extra free volume to polymer matrix. However, when the aliphatic chain is too short, it is usually volatile, while when it is too long, the processability of the polymer blend might be difficult. Aromatic groups usually act as compatibilizer blocks. Most of the commonly used plasticizers contain an aromatic ring in their structures, e.g., phthalates, terephthalates and trimellitates. The presence of an aromatic ring introduces great flexibility to the plastics. Plasticizers with ester groups are very common due to their interactions with polymers based on the van der Waals forces, hydrogen bonds and electrostatic interactions. Those plasticizers act as cohesive structures, avoiding leaching and migration of the plasticizer from the plastics. It should be emphasized that usually there is the same number of ester groups and aliphatic chains in a plasticizer molecule (phthalates, trimellitates) [8,17].

### 1.3. Migration of Plasticizers

Under given conditions, the permanence of a plasticizer in a flexible polymer product depends on several factors, namely its structure, chemical composition, molecular weight, and polarity. Because a plasticizer molecule is usually not chemically bonded to a polymer chain, it can be released during manufacturing of a polymer or, later, during its everyday use [27]. Migration of plasticizers from the plastics during the product lifetime deteriorates the physical properties of the material and can cause the contamination of surroundings [28]. Phthalate esters, particularly, are known to be the most dangerous plasticizers for humans and the natural environment. They can be found in soils [6,29], food products [30,31,32], human breast milk [7,33], indoor air [34], bottled water [35,36], cosmetics [37], etc. Because of their lipophilic properties, phthalates can accumulate in lipids and cause serious damage, e.g., to human liver [38,39]. Studies describing the impact of phthalates on human health have shown that they seriously influence the production of hormones and cause endocrine disturbances, which can entail considerable risks of autism, endometriosis, asthma, and difficulties in pregnancy [28,40]. More information on the biomonitoring of phthalate exposure can be found in the recent work of Wang at al. [41] Therefore, the use of several plasticizers, including buthyl benzyl phthalate (BBP), di(n-butyl) phthalate (DBP), bis(2-ethylhexyl) phthalate (DEHP), diisodecyl phthalate (DIDP), diisononyl phthalate (DINP) and di-n-octyl phthalate (DOP) is regulated by the European Union and some other countries. The directive 2005/84/EC of the European Parliament and of the Council of 14 December 2005 restricts using DEHP, DBP and BBP in all toys and childcare products, and using DINP, DIDP and DOP in children’s items that can be taken into the mouth [42]. Biodegradation of phthalates is highly important in order not to increase their quantities in the environment. Phthalates can be degraded by microorganisms and in this way can be eliminated from the environment, and the most effective disposal approach is through hydrolytic decomposition to phthalic acid and alcohols [43].

The migration studies of plasticizers are usually divided into four types:Volatility;Extraction and leaching;Migration into solids;Exudation under pressure.

The term “migration stability” is often used in each of the aforementioned phenomena. It is usually described as a change in weight after exposure to a particular agent: air, liquids, or solids. The weight loss always depends on the plasticizer content in the PVC, as well as all media that are in contact with the plastics. Obviously, both thermal and pressure conditions play an important role in determining the migration of plasticizers [27,44].

### 1.4. General Strategies to Replace Conventional Plasticizers

In the last few years researchers were trying to find a plasticizer which could replace petroleum-based phthalates including DEHP, the most popular plasticizer. One of the first attempts was to search for compounds that have similar structures and physicochemical properties to commonly used plasticizers. In this case, compounds based on furandicarboxylic acid were proposed as novel plasticizers [45,46,47]. In 2004, 2,5-furandicarboxylic acid was chosen as one of twelve bio-based building block chemicals by the U.S. Department of Energy, mainly because of its availability to substitute terephthalic acid, i.e., in polyesters. Because furandicarboxylic acid has a lower molecular weight (156 g/mol) than phthalic acid (166 g/mol), its esters exhibit a higher volatility compared with the esters in phthalic acid. Therefore, to lower the volatility of a potential plasticizer, alcohols with longer chains should be introduced during the esterification reaction [46].

To find alternative plasticizers to phthalates, researchers are searching for more branched plasticizers than conventional ones. The new plasticizers, particularly highly branched esters and polyesters exhibit improved migration resistance and reduced volatility due to higher molecular weight. Besides, they exhibit a three-dimensional network, as well as an ability to entangle with PVC chains. In comparison to their linear counterparts, hyperbranched plasticizers exhibit high terminal functional group concentrations and, therefore, are able to increase free volume in polymer matrices. On the other hand, a high molecular weight plasticizer is usually difficult to process, especially because of its higher viscosity [48,49,50,51]. 

In the last few years, two strategies have been introduced for the replacement of phthalates, namely total or partial replacement of a plasticizers (Figure 1). The pursuit of new compounds was accompanied with numerous tests that were carried out to investigate the physical and chemical properties of PVC blends, as well as toxicological studies of the plasticizers [52]. When looking for new plasticizers, research has focused on lowering the glass transition temperature (which means that a plasticizer has better plasticization efficiency), associated with an increase in elongation at break value and a decrease in the tensile strength of material. Furthermore, migration resistance, including volatility, exudation and extraction, is an important factor in developing new compounds for PVC formulations. There are two main groups of plasticizers: primary—the main plasticizer in a PVC blend, added in large quantities and highly compatible with PVC; and secondary—having limited compatibility with PVC, easily migrating, especially when used in excess and usually added to the blend to improve its properties or to reduce its costs [53].

The introduction of a secondary plasticizer to a PVC blend can result in the presence of synergistic interactions between the blend components that could influence the properties of the final product. There are several reports describing the situation when a part of the primary plasticizer was replaced by a secondary plasticizer [54,55,56]. Due to the interactions between these two compounds, the plastics could adopt better properties, particularly good migration resistance and enhanced mechanical properties compared to a material with only the primary plasticizer. Primary plasticizers are often small molecules and are used because of their good plasticization efficiency. Nevertheless, they have a distinct tendency to migrate because of their high volatility. The intermediate effect of this in most cases is obtained when a high molecular weight secondary plasticizer, usually polymeric macromolecule with an excellent migration resistance, is added to the mixture, partially replacing the primary one [56,57,58]. Because of its higher molecular weight (according to Wypych a polymeric plasticizer should have an average MW > 2000 g/mol [18]), the plasticizer migration from the plastics is reduced.

Interestingly, when two or more plasticizers are used in a polymer system, a phenomenon of co-solvency may appear. The synergistic effects may be noticed, particularly when one of them is a relatively small molecule and the second is a macromolecule, e.g., a mixture of branched phthalates and octyl-diphenyl phosphate. In 2003, van Oosterhout presented the study of compatibility of PVC-plasticizers when binary and ternary blends of plasticizers were introduced [59]. Solid-gel transition temperatures (known also as a clear point) and Flory-Huggins interaction parameter χ were determined for several blends containing linear and branched phthalates, aliphatic dicarboxylic esters, trimellitate, phosphate, alkyl sulfonic ester of phenol and polymeric plasticizers poly(butylene adipates) with varying molecular weights. In this study, interaction coefficients were predicted and some synergistic effects were observed when two or more plasticizers were used in PVC mixture, e.g., higher solvent power than expected for a mixture of octyl-diphenyl phosphate with one or two branched phthalates increasing with the chain length of the phthalate compound. Such synergy was not observed when three phthalates were mixed with octyl-diphenyl phosphate. A synergistic interaction between dioctyl phthalate and polymeric plasticizer poly(butylene adipate) for clear temperature values was also observed.

Plasticizing theories indicate that plasticizers do not create covalent bonds with PVC. Still, there are numerous studies mentioning reactive plasticizers, which are compounds able to attach to PVC chains. Two extensive reviews describing internal plasticizers were written by Bodaghi [60] and by Ma et al. [52] These compounds are usually based on commonly used plasticizers like DOP [61] and triethyl citrate [62]. Bodaghi proposed two main types of reactive plasticizers depending on the type of the reaction of a plasticizer with the polymer chain: nucleophilic substitution reaction;click reaction.

As expected, the internal plasticizers exhibited zero migration, but provided relatively high T_g_ values of PVC blends, the latter one indicating their worse plasticizing efficiency [21,63].

## 2. New Primary Plasticizers

### 2.1. Branched and Hyperbranched Compounds

#### 2.1.1. Branched and Hyperbranched Compounds Derived from Adypic Acid

An interesting comparative study of the performance of linear and branched polymeric plasticizers in comparison with DOP and DEHP was carried out by Lindström and Hakkarainen [21,48]. The polymeric plasticizers, poly(butylene adipate)s (PBAs), were obtained using different ratios of adipic acid or dimethyl esters of adipic acid and 1,4-butanediol. Molecular weights of the obtained polyesters varied from 2000 to 10,000 g/mol, and these compounds formed a semi-miscible two-phase system with PVC. An excellent miscibility of plasticizer and PVC matrix is provided by the hydrogen bonds between the carbonyl group of the ester and α-hydrogen of PVC chain. The best hydrolysis-resistant properties were exhibited by PVC/PBA films with branched polyesters synthesized from dimethyl ester of adipic acid in which the amount of monomeric degradation products (adipic acid, butanediol) was close to zero. The results have shown that methyl ester end-groups prevent polyester hydrolysis and branched polyesters are characterized by the lowest water absorption. Moreover, PVC films with branched plasticizers exhibited improved mechanical properties (higher strain at break) than linear films, however only three films with branched polyesters exhibited a higher strain at break than films with DOP and DEHP. Unfortunately, migration studies of these plasticizers were not conducted. 

In 2011, Ascione et al. [64] synthesized hyperbranched poly(butylene adipate) (HPBA) and applied it as a plasticizer to PVC with different plasticizer concentrations (50, 60, 70, 80 phr). The properties of as-formed blends were compared to a blend with linear poly(butylene adipate) (LPBA) (50, 60, 70, 80 phr) and with DOP (70 phr). The best migration resistance in an exudation test was exhibited by HPBA (70 phr), which was explained by entanglement of the polyester with PVC chains that made it harder to migrate. As expected, DOP presented the highest migration; however, this polymer–plasticizer blend was characterized by the lowest T_g_ value (7 °C), which pointed to a better plasticization efficiency. In contrast, HPBA blended with the same plasticizer content exhibited T_g_ of 23 °C.

#### 2.1.2. ɛ-Caprolactone Based Branched and Hyperbranched Compounds

Polymeric plasticizers based on poly(ɛ-caprolactone) were described in the literature as plasticizers that could replace toxic phthalates in PVC formulations. These plasticizers are biocompatible, nontoxic, exhibit low T_g_ values (only a few degrees higher that blends with phthalate plasticizers) and a good migration resistance [19,51,65].

For instance, poly(ɛ-caprolactone) (PCL) presents good compatibility with PVC, so it can be successfully applied as a plasticizer macromolecule. In 2018, Huang et al. [23] developed hyperbranched PCL with different molecular weights (varying from 8100 to 59,000 Da) through the polymerization of ɛ-caprolactone initiated by glycidol, followed by the copolymerization reaction of glycidol functionalized PCL (GPCL) and succinic anhydride (SA). The number of consecutive copolymerization steps of GPCL and SA was called a generation and denoted as D_n_, where n is the generation number. The comparison of new plasticizers with DEHP revealed their slightly higher T_g_ (particularly for first generation D_1_ plasticizer), and their higher ultimate elongation (for 10–50 phr of D_1_ plasticizer). The tensile strength was very similar for all the blends with D_1_ independently from the plasticizer content, but was lower than for DEHP blends with 10, 20 and 30 phr of plasticizers. For higher plasticizer contents, blends with a D_1_ plasticizer exhibited a higher tensile strength. The studied plasticizers exhibited excellent migration resistance, which was close to zero during volatility and extraction tests in n-hexane. As expected, the higher the molecular weight of a plasticizer (which related to the higher generation of polymers), the lower the weight loss in migration tests. From all tested polymeric plasticizers, the most advantageous properties presented the plasticizer of first generation. Plasticizers whose molecular weight and degree of branching were too high exhibited worse plasticization properties because of limitation in PVC backbone mobility due to the specific interactions between the carbonyl groups of the polymeric plasticizer and the chlorine groups of the PVC chain. The existed donor–acceptor interactions between C=O and CH-Cl groups resulting in good plasticizing efficiency and excellent migration resistance in case of the D_1_ generation [23].

Similar studies were carried out by Li et al. [50] The new plasticizer was synthesized through a reaction of glycidyl and succinic anhydride. In the second step, the obtained hyperbranched polyester acted as an initiator in reactions with ɛ-caprolactone via ring-opening polymerization. Hyperbranched block copolymer polyester-b-poly(ɛ-caprolactone) with different molecular weight values, ɛ-caprolactone: hyperbranched polyester ratios and generations of polymer were obtained. Here, the generation of polymers was understood as a number of caprolactone units in each arm of a star-shaped polymer. The molecular weight values of the new polymeric plasticizers were lower than those obtained by Huang et al. [23] and varied from 6158 to 14,872 g/mol. A PVC blend with synthesised hyperbranched plasticizers with the highest ratio (15:1) of caprolactone: polyester (named HEPCL_8_) exhibited the lowest T_g_, even in comparison with DOP. Elongation at break values of PVC films with DOP and different hyperbranched plasticizers were comparable and the tensile strength was slightly higher for new plasticizers than for DOP. The new compounds were characterized with a good thermal stability and improved plasticizer migration resistance. Authors attributed the observed improvement in thermal stability to the presence of dipole–dipol interactions between the polar ester group of plasticizer and chlorine atoms of PVC that reduces the electronegativity of the chlorine atoms. The studies showed that only a PVC blend with HEPCL_8_ performed better than DOP in terms of mechanical properties, and at the same time exhibited the highest migration resistance.

#### 2.1.3. Other Branched and Hyperbranched Plasticizers

Another hyperbranched compound that can be used as a plasticizer is alkyl terminal hyperbranched polyglycerol (alkyl-HPG) synthesized from different molar ratios of glycidol and trimethylolpropane [49]. Those hyperbranched polyglycerols with terminal hydroxyl groups could react with butyric anhydride, acetic anhydride or octanoyl chloride to obtain terminal alkyl chains with a different number of carbons. In this way, five new plasticizers were obtained with the general formula HPGx-Cy (“x” refers to molar ratio of glycidol to trimethylolpropane and “y” refers to number of carbon atoms in a terminal alkyl chain) named: HPG3-C4, HPG9-C2, HPG9-C4, HPG9-C8, HPG21-C4. The results showed that the compounds with too-short (HPG9-C2; only two carbon terminal alkyl chain) and too-long alkyl chains (HPG9-C8; eight carbon terminal alkyl chain) were not miscible with PVC. The number average molecular weights of the synthesized compounds varied from 1571 to 2504 g/mol. Comparison with DEHP showed that PVC films plasticized by HPGs exhibited a slightly worse plasticization efficiency, namely higher T_g_ (from −18.4 to −14.4 °C) and a lower elongation at break. Thermal stability studies showed that PVC blends with HPGs were more stable (exhibiting higher onset temperature values on TGA curves) than samples with DEHP. A great number of ether and ester groups in a plasticizer molecule provide donor-acceptor interactions with PVC, resulting in an excellent migration resistance. As expected, the migration resistance was significantly higher than in films plasticized by DEHP because of the higher molecular weight of alkyl-HPGs [49]. 

The apposite balance between the polar and non-polar segments of a plasticizer molecule determined its compatibility with polymer matrix and plasticization effectiveness. In 2020, V.A. Pereira et al. [66] compared end-capped saturated polyesters synthesized from phthalic anhydride, sebacic acid and different ratios of 1,3-propanediol: alkyl chains (branched 2-tetradecyloctadecan-1-ol (TDOD) or unbranched octanol), with analogues polyester without end-capped units/groups (named SP00). The authors hypothesized that by endcapping investigated saturated esters with selected long alkyl chains alcohols, it would be possible to improve the interactions with PVC and ensure enhanced plasticizing effects due to the change in a polar/non-polar balance of the plasticizer. The lowest T_g_ value was obtained when polyester with octanol was added to the mixture. Higher molecular weights (from 2200 to 6100 g/mol) entailed difficulties in a movement of the chains, which could be the reason for the observed higher T_g_ values in comparison with DOP. The highest elongation at break value was exhibited by the blend with the SP00 plasticizer. In the case of the blend with SP04, it exhibited only slightly lower elongation than DOP, but the elongation at break of blends with plasticizers synthesized from 2-tetradecyloctadecan-1-ol was significantly lower. Compounds with alkyl chains also exhibited better thermal stability and migration resistance in comparison to DOP. This study clearly showed the superiority of the unbranched chain. However, to understand the differences in properties of these two types of plasticizers, a study using shorter chains in the branched unit should be conducted, i.e., 2-octylodecan-1-ol or branched alcohol with the same number of carbon atoms. 

### 2.2. Vegetable Oil-Based Plasticizers

In the last few years, researchers have focused on finding new plasticizers from renewable resources [67,68,69]. In this case, nontoxic vegetable oils were often used, e.g., soybean oil [70,71,72], sunflower oil [73,74], castor oil [75,76,77,78], rice bran oil [79], tung oil [80] etc. These compounds are environmentally friendly and do not cause adverse effects to human health. Since bio-based plasticizers originate from renewable resources, they are usually biodegradable. The main disadvantage of these natural materials is their relatively high cost of production, which is significantly higher than for DOP. Therefore, the current challenge is to find an easy-to-process plasticizer with production costs comparable with the most common plasticizers [81]. Another issue is the limited compatibility of vegetable oils with PVC that makes it necessary to modify them before use. An important aspect of the production of bio-plasticizers is that these materials are food for some animals and humans, so in some cases it could disturb the balance in the ecosystem [82]. Unsaturated bonds of natural oils in most cases are epoxidized to obtain epoxy groups that are able to scavenge hydrogen chloride released during thermal degradation, so this treatment increases the thermal stability of plastics [83]. Such plasticizers usually exhibit relatively high molecular weights and therefore have good migration resistance.

The most common bio-based plasticizer is epoxidized soybean oil (ESBO) [24,25,84,85], which is also used as a heat stabilizer in PVC blends because of its ability to scavenge HCl molecules released during heating and is responsible for the propagation of thermal degradation of the polymer [86]. When ESBO is used as a plasticizer, a content of 25–45 wt% is usually needed to prepare the plastics. Adding ESBO as a stabilizer is limited to 1–2 wt% [66,87].

In 2019, Tan et al. [24] obtained dimer and trimer acid from soybean oil and used it in the esterification reaction with polyethylene glycol methyl ether. The impact of a number of oxethyl units in dimer acid was studied, because of its ability to create hydrogen bonds with a PVC matrix, which implies good plasticizing efficiency. The results showed an excellent compatibility with PVC for plasticizers containing six (DA-6) and eight (DA-8) oxethyl units. The DA-8 plasticizer blend with PVC presented the best tensile properties in comparison with DOTP: comparable elongation at break and tensile strength. A volatility test showed lower weight loss of the new plasticizers than those observed for DOTP, which is obvious due to the higher molecular weight of those novel plasticizers. The studied plasticizers in most cases exhibited leaching resistance like DOTP; however, in hexane this resistance was much better than for DOTP. This observation could be explained by the strong non-polar behaviour of hexane and the presence of many slightly polar groups in the new plasticizer, namely ether and ester groups.

A series of palm oil-based plasticizers prepared using palm oil, glycerol, phthalic anhydride, acetic acid, and calcium hydroxide in different compositions was proposed by Rozaki et al. in 2017 [86]. These plasticizers exhibited varying molecular weights (from 2279 to 9560 g/mol) and caused a decrease in T_g_ when compared with neat PVC, but was not as significant as in the case of DEHP or triethyl citrate. Leaching resistance was performed using two methods: UV-Vis and weight methods, in a n-hexane and an ethanol/water solution. Although the UV-Vis method is not very common in studying plasticizer migration, a height a peak at 199 nm attributed to ester groups could provide the information about the occurrence of leaching. Both methods indicated DEHP as the plasticizer with the highest migration, and palm oil-based as the plasticizer with the highest molecular weight as the one with the lowest migration. Both the tensile strength and elongation at break were improved when the palm oil-based plasticizers were used to plasticize PVC, and the mechanical properties of the PVC blend were much better than those in case of DEHP. A good performance of these plasticizers should be attributed to the interactions between plasticizer molecules and PVC chains that link them together. Authors postulated that the possible interactions between carbonyl, as well as hydroxyl groups of the plasticizer and the carbon-chlorine of PVC were responsible for good plasticization efficiency.

Moreover, an oleic acid-based plasticizer was described by Omrani et al. [88] It was obtained using reactions with thioglycolic acid and then with hydrogen peroxide to get sulfone group. In the last step, esterification of the obtained diacid (DA) using different alcohols—methanol (obtaining MDA) or ethanol (obtaining EDA)—was carried out. The molecular weight of both MDA and EDA (434 g/mol and 462 g/mol, respectively) were slightly higher than the molecular weight of DOP (390 g/mol). The authors evidenced reduced leaching in different solvents, volatilization, and exudation of plasticizers from PVC blends in comparison to a blend with DOP. The leaching resistance was slightly better for the methyl ester than for the ethyl ester, which could be associated with the stronger interaction of MDA derived from its higher polarity. This difference was also manifested in the inversed results obtained in volatilization and exudation tests and was associated with the higher molecular weight of EDA. The tensile strength and elongation at break studies showed no significant differences between new plasticizers and DOP. Besides, new plasticizers presented a lower viscosity than DOP, which is very important in PVC plastics production. Furthermore, the thermal stability of PVC blended with MDA or EDA was higher than for DOP, so the studied oleic acid-based plasticizers could serve as a promising alternative to phthalate plasticizers.

In 2014, Vieira et al. [89] synthesized a plasticizer based on rice fatty acid obtained in a polyesterification reaction with polyols. The mechanical properties of a PVC blend were compared only to neat PVC and, as expected, the tensile strength was lower, and elongation was higher for the plasticized material than for neat PVC. Nevertheless, to evaluate the plasticizing efficiency of the plasticizer, a comparison with a commercially used plasticizer should be investigated, as well as with migration resistance studies.

Moreover, cardanol derivatives have been widely described as potential plasticizers [82,90]. Similar to fatty acids, their unsaturated bonds are usually epoxidized to increase their compatibility with PVC. In 2018, Greco et al. [82] proposed a method of cardanol modification to obtain epoxidized cardanol acetate (ECA) with essential use of the eco-friendly reagents instead of m-chloroperbenzoic acid and dichloromethane. In this case, a lipase catalyst was used to catalyze the epoxidation reaction: fresh lipase for obtaining the CDPac-ep1 and reused lipase to obtain CDPac-ep2 and CDPac-ep2r compounds. The tensile strength of the PVC material plasticized with 70 phr of cardanol derivative was similar to the material with 55 phr of DEHP; however, after ageing, the tensile strength distinctly increased, so even in comparison to PVC plasticized with 70 phr of DEHP, the plasticizing efficiency of the cardanol plasticizer was very poor, which may suggest the presence of some undesirable crosslinking processes. Furthermore, the migration resistance was lower for the cardanol derivative, causing significant deterioration in mechanical properties after aging.

### 2.3. Other Primary Plasticizers 

Several studies on the modification of plasticizer molecules were conducted to reduce the migration of the plasticizer from plastics. In 2020, an interesting family of bio-based plasticizers were studied by Nguyen et al. [91], who synthesized furan dicarboxylate-based plasticizers using galactaric acid from marine biomass and several bioalcohols (1-butanol and 3-methyl-1-butanol also known as isoamyl alcohol). Because of an extra oxygen atom in the heteroaromatic ring of furan, investigated plasticizers exhibited a stronger interaction with PVC than commonly used DOP plasticizer. DSC studies showed that a blend of PVC with furan dicarboxylates reacted with 1-butanol exhibited similar T_g_ values as PVC/DOP, especially when the plasticizer content was higher than 42,9 phr. T_g_ values of plasticizers with isoamyl alcohol, providing branched alkyl chain in the molecule, were significantly higher than for plasticizers with a linear chain. Volatility studies showed significantly higher weight loss of all new plasticizers from PVC material because of their low molecular weight (268 g/mol for isomers with 1-butanol and 296 g/mol for isomers with isoamyl alcohol, respectively). The leaching resistance of furan dicarboxylates determined in hot hexane was higher than for DOP, which was associated with higher polarity of dicarboxylates caused by the oxygen atom in the furan ring.

A promising modification seemed to be a hydrosilylation reaction carried out on plasticizer molecules, since Si-O linkage exhibits relatively weak basicity, resulting in chemical inertness, and free internal rotation around the bond in comparison to C-O [92,93]. Siloxanes exhibit also good thermal stability, hydrophobicity, and nontoxicity. Consequently, Ji et al. [22] reported the use of poly(maleic acid hexanediol) ester (MH) modified with 10% (MHA-1) and 20% (MHA-2) of the maleic anhydride amount with 1,1,3,3-tetramethyldisiloxane (TMDS). The hydrosilylation reaction was carried out with the use of an unsaturated bond of maleic acid. Thus, the obtained plasticizer was introduced in a PVC blend, and several properties of these plastics were investigated. The elongation at break of the blend with MHA-1 was the highest in comparison to blends with DOP, MH and MHA-2. As expected, the thermal stability of PVC blends with MHA-1 and MHA-2 was higher than of PVC blends with MH. Furthermore, the migration resistance, including extraction, volatilization, and exudation of the polymeric plasticizers was higher because of their higher molecular weight. This enhancement in the characteristics of PVC blends with the proposed plasticizers could be also associated with a new hydrogen bond formed between oxygen of the Si-O-Si group and α-hydrogen of the PVC chain. However, T_g_ values (from −11.18 to −17.51 °C), and hence the plasticizing efficiency of the studied plasticizers, was slightly lower than for DOP (T_g_ of −22.05 °C).

Chlorinated products, especially chlorinated paraffins, are widely used in PVC formulations as secondary plasticizers and flame retardants, and reduce the manufacturing costs [94,95]. Under specific conditions, these compounds could be also used as primary plasticizers. In 2017, Yuan and Cheng [96] covalently attached a chlorinated paraffin molecule to DOP using 4-hydroxyphthalic acid and introduced the new plasticizer into a PVC blend. In this case, both carbonyl groups and chlorine substituents of chlorinated paraffin were involved in the interaction with PVC. The extraction in heptane studies showed no weight loss for a PVC blend with a new plasticizer. T_g_ values of PVC blends with the modified plasticizer and DOP were comparable. Although seemingly promising, the applicability of this new plasticizer should be further verified by the characterization of mechanical properties of a PVC blend.

In 2017, Miao et al. [26] introduced poly(nadic anhydride) polyesters as a new plasticizer in PVC films to substitute DEHP. A series of potential plasticizers was synthesized from nadic anhydride and different polyols: 2-methyl-1,3-propanediol (P-MPO-NA), 1,4-butanediol (P-BDO-NA) and diethylene glycol (P-DEG-NA). The FTIR studies showed the broadening and shifting of the peak ascribed to the C-O–C stretching vibration due to the increase in electron density of the -C-O-C=O group, revealing the strong complexing interaction between P-NAs plasticizers and PVC. As expected, due to a relatively high molecular weight (about 9000 Da), all three new plasticizers exhibited better migration resistance from the PVC blend than from DEHP. The highest elongation at break and the lowest T_g_ was found for PVC/P-MPO-NA film, probably because of the fact that MPO was the shortest and most branched alcohol used in the study, confirming the positive influence of branched chains in a plasticizer molecule on plasticizing efficiency. 

In 2018, Brostow et al. [84] compared the properties of toxic (ditridecyl phthalate and trinonyl benzene-1,2,4-tricarboxylate) and nontoxic (based on hydrogenated castor oil and ESBO) plasticizers. In this study, a mixture of two plasticizers (hydrogenated castor oil and ESBO) formed a composed system (1:1 wt. ratio) with a synergistic effect on plastic properties. As evaluated during exudation studies in different temperature values ranging from room temperature to 136 °C, in each temperature (except for room temperature) the weight loss of the samples with one plasticizer was higher than in a sample with a mixture of both of them. It is worth saying that at 136 °C, the weight loss of the sample with hydrogenated castor oil was approximately four times higher than in a sample with the mixture of two plasticizers. Moreover, the elongation at break increased when the two plasticizers were mixed in a PVC blend. This could be explained by the possible interactions between hydroxyl groups of hydrogenated castor oil and epoxy rings of ESBO [84]. In this study, ESBO probably served as a heat stabilizer, but the TGA analysis was not investigated for a sample with pure hydrogenated castor oil, so the effect of ESBO on thermal stability cannot be estimated.

The available experimental results showing the performance of PVC plasticized with new primary plasticizers described in this section are collected in Table 1, which contains, if available, mechanical properties, glass transition temperatures and the results of different migration tests that were conducted in cited works. All these data are compared to the “standard blend”, i.e., blends with a common commercial plasticizer that usually was DEHP or DOP. In some cases, like in [89] and [84], rare reference standards are provided, i.e., pure non-plasticized PVC and a mixture of DTDP/TIMTM, respectively. The correlation between elongation at break and tensile strength, and the differences in glass transition temperatures of all the discussed plasticizers rationalized to the reference values received for blends with commercial plasticizers are presented in Figure 2 and Figure 3, respectively.

## 3. New Secondary Plasticizers

In the last few years, a trend in the research could be noticed to induce synergistic effects on plastic properties by the application of secondary plasticizers. Although these materials are not always less compatible and inexpensive, they can significantly influence some desired properties of plastics. In 2018, Lee et al. [25] reported the synergistic effects of cardanol-based plasticizers (cardanol acetate (AC) and epoxidized cardanol acetate (ECA) mixed with epoxidized soybean oil (ESBO). PVC blends were prepared by the addition of 30 or 50 phr of plasticizer to PVC dissolved in tetrahydrofuran, and some formulations were further supplemented with 5 phr of ESBO. To compare obtained results, samples with a commonly used plasticizer, DOP, were prepared. PVC with ECA and ESBO exhibited a higher elongation than PVC without ESBO. When ESBO was added to DOP, the elongation of the PVC sample was lower than for the PVC blend with only DOP (50 phr). In this way, a synergistic effect of ECA and ESBO was clearly presented. Furthermore, the FTIR studies of the mixture of ECA and ESBO showed that using these plasticizers together resulted in a decrease in the ratio of crystalline/amorphous bands. Based on the most prominent decrease in the intensity of amorphous bands at 1435 and 610 cm^−1^ observed for the PVC/ECA 50 phr blend, it can be concluded that ECA efficiently reduces the chain-to-chain interactions in PVC, outperforming the other studied plasticizers. Leaching studies in deionised water showed that PVC/DOP exhibited the highest weight loss (12.9%) among the investigated plasticizers, whereas the weight loss of ESBO was close to zero, as explained by its high molecular weight (approximately 1000 g/mol). Although molecular weights of cardanol-based plasticizers were lower than DOP, these plasticizers presented enhanced leaching resistance. Unluckily, the mixture of plasticizers was not studied in terms of leaching resistance.

In 2004, Sunny et al. [97] partially replaced DEHP by three different polymeric plasticizers: nitrile rubber NBR (medium acrylonitrile content), carboxylated nitrile rubber XNBR (7.5% carboxylation) and epoxidized natural rubber ENR (50% epoxidization). The addition of up to 25 phr of XNBR or ENR to PVC provided lower a tensile strength than for pure DEHP. Using ENR, no significant changes were observed in tensile strength. Elongation at break in all samples with polymeric plasticizers, regardless the primary to secondary plasticizer ratio, was lower than with pure DEHP. Leaching tests evaluated using a petroleum ether showed the decrease in DEHP leaching for all samples with a mixture of plasticizers when compared with pure DEHP, but there were no distinct and advantageous synergistic effects observed, assuming that migration of polymeric plasticizers was close to zero. Unfortunately, the study on the migration of pure polymeric plasticizers from plastics was not performed.

In 2017, Wang et al. [57] developed a new plasticizer containing hydroxyl and nitrogen groups, synthesized by a reaction of melamine and formaldehyde, followed by a reaction with tung-maleic anhydride and then with epichlorohydrin. Tung-maleic anhydride was added in four different molar ratios: 1:1 (GEHTMA-1), 1:2 (GEHTMA-2), 1:3 (GEHTMA-3) and 1:4 (GEHTMA-4). Only PVC with GEHTMA-3 exhibited higher elongation and tensile strength in comparison to DOTP, so this compound was applied in plasticizer mixtures. Studies on samples with different DOTP/GEHTMA-3 ratios were prepared to investigate the synergistic influence of these two compounds on migration resistance and mechanical properties. Because of the presence of different functional groups in GEHTMA-3 macromolecule, i.e., ester, hydroxyl, epoxy, benzene ring, etc., the newly developed secondary plasticizer was able to strongly interact with both DOTP and PVC, forming hydrogen bonds and dipole–dipole exchange. A significant decrease in weight loss during extraction in petroleum ether and n-hexane was found in all samples, even in the samples with a relatively small content of tung oil-based plasticizer (4:36 phr of GEHTMA-3: DOTP). Therefore, GEHTMA-3 containing ester groups, hydroxyl groups, epoxy rings and nitrogen, was shown to act as a compatibilizer between PVC and DOTP. Elongation at break was improved when 4, 8, 12, 20 phr of GEHTMA-3 was added, and tensile strength of samples with plasticizer mixtures was slightly higher in comparison to pure DOTP (22.39 MPa for pure DOTP and 27.89 to 33.13 MPa for the mixture, respectively).

In 2020, Chen et al. [56] synthesized a hyperbranched ester plasticizer using soybean oil (SOHE) and mixed it with PVC for studying thermal and mechanical properties. The new compound exhibited good thermal stability in comparison to DOP, but T_g_ was significantly higher (61.89 °C for SOHE and 41.46 °C for DOP, respectively). However, a mixture of those two plasticizers, when DOP was partially replaced by SOHE, exhibited only slightly higher T_g_ than pure DOP. Moreover, the mechanical properties were improved when a mixture of plasticizers was applied in PVC blends, whereas a blend with pure SOHE showed worse mechanical properties (lower elongation at break and higher tensile strength) than DOP. Authors stated that the interaction between the polar ester groups of SOHE and α-hydrogen atoms of PVC results in the formation of hydrogen bonds, a weakening of PVC chain-to-chain attractions and an increase in the free volume. Accordingly, a synergistic effect could be easily seen when a mixture of plasticizers was added, making SOHE a promising secondary plasticizer. As expected, the extraction and volatility resistance in blends with SOHE was improved due to the high molecular weight of the studied plasticizer.

The studies of sunflower oil were undertaken by Rouane et al. [55] to provide a new plasticizer using an epoxidation reaction. PVC samples with partial substitution of DEHP were investigated in different epoxidized sunflower oil (ESO)/DEHP ratios. Tensile strength of PVC plastics with ESO was lower than with pure DEHP; however, elongation at break decreased with the increasing ESO content. It was concluded that the presence of ESO induced a deteriorating effect on the mechanical properties of PVC, making it more brittle and rigid. Therefore, the amount of ESO in the plasticizer mixture (total amount 60 phr) should be kept up to 20 phr. UV-Vis and FTIR analysis of prepared plastics show that the addition of ESO as a secondary plasticizer to the PVC blend with DEHP provided a good thermal stability. This phenomenon was explained by the ability of epoxy groups present in ESO to scavenge HCl molecules released from PVC. Moreover, the studies of migration resistance of new formulations were not conducted.

An interesting application of cardanol derivatives as a secondary plasticizer was proposed by Chen et al. [54], who investigated a synergistic effect of cardanol acetate (CA) and DOTP. Because of a relatively high functional group concentration in relation to molecular wieght, especially in blends with epoxidised CA, it exhibited good plasticizing efficiency, mainly due to the effective reduction of chain-to-chain interactions in PVC. The obtained values of tensile strength were lower and elongation at break were significantly higher in PVC blends with a mixture of DOTP and CA than when pure DOTP was used. Furtheromore, the T_g_ values of PVC plastics were lower when DOTP was partially replaced by cardanol acetate, indicating better plasticizing efficiency. Unfortunately, the migration resistance of blends with mixtures of plasticizers was not investigated. Nevertheless, plasticizer migration from PVC blends with only one plasticizer, including extraction in distilled water, exudation and volatility, was investigated, showing the highest volatility of PVC/CA (about three times higher than for PVC/DOTP) and comparable extraction in distilled water. Still, a study of migration resistance should be carried out to investigate whether the synergistic effect on the migration of two plasticizers occurs. Noteworthy research focused on the introduction of a low molecular weight plasticizer as a secondary plasticizer and was carried out by Matos et al. [47] DEHT was partially (5, 10, 15, 20 phr) replaced by di(ethylhexyl)-2,5-furandicarboxylate (DEHF) in PVC plastics (with whole plasticizer content of 55 phr). When the main plasticizer was replaced by 5 and 10 phr of DEHF, the elongation at break of the PVC sample was significantly elevated; however, a higher content of the secondary plasticizer caused a decrease in elongation. As in previous studies, furan-based plasticizer was characterized by higher weight losses in migration studies, especially in volatile resistance tests. In leaching studies in hexane, similar results as Nguyen et al. [91] were collected. The results indicated that furan-based plasticizers could be used, i.e., in materials that are in contact with food with a high fat content, because of their relatively high polarity. An FTIR study revealed that at the relatively low contents of DEHF in the DEHF/DEHT plasticizer mixture (lower than 10/45 phr), it acted as a secondary plasticizer and dominated the interactions between PVC and DEHT. However, in blends with a content of DEHF higher than 10 phr, it started to play the dominant role, and its interactions with PVC restricted the migration of the plasticizer. 

To solve the problem of the high costs of bio-based raw materials used in the production of plasticizers, the use of waste cooking oil as a bio source was proposed by Zheng et al. and others [81,98,99]. Waste cooking oil 2-ethylhexyl esters (WCOEtHEs) were obtained from waste cooking oil methyl esters (WCOMEs). The epoxidation reaction was also carried out using both, WCOEtHEs and WCOMEs. Characterization of PVC blends with the new plasticizers was carried out when DOP was partially replaced by the waste cooking oil-based plasticizer. Studies of migration stability and extraction resistance showed significantly higher weight loss of PVC/epoxidized WCOEtHEs blends than the PVC/DOP + epoxidized WCOEtHEs, especially in kerosene. The high content of the polar epoxy groups in epoxidized Ep-WCOEtHEs probably held the cross-link density, thus reducing the chain mobility between polymer molecules. Usually, when tensile strength increases, the elongation at break decreases, so it is hard to develop a plasticizer that has both parameters improved, especially when macromolecules are used as plasticizers. In this case, however, elongation at break increased, and tensile strength decreased when two plasticizers were incorporated in a PVC mixture in comparison to blends with pure DOP.

The experimental results showing the performance of PVC plasticized compounds described in this section are collected in Table 2, which contains information about the examined primary and secondary plasticizers, their content and ration in PVC blends and, if available, several results of mechanical tests, thermal properties (glass transition temperature) and stability in the sense of plasticizer migration. The correlation between elongation at break and tensile strength, and the differences in glass transition temperatures between all discussed mixed plasticization systems contained secondary plasticizers rationalized to the reference values received for PVC blends with only the same primary plasticizers are presented in Figure 4 and Figure 5.

## 4. Internal Plasticizers

Covalent bonding of plasticizer molecules and PVC chains is an effective way to suppress the migration of a plasticizer from plastics. Although they not able to exhibit good plasticization efficiency, new internal plasticizers are still developed. There are several studies describing internal plasticizers and this issue is still being investigated. Plastics made of such compounds could potentially find an application in demanding markets, such as automotive, food packaging, etc.

In 2017, Jia et al. [100] elaborated on the synthesis of new internal plasticizers and investigated their properties. Since esters of citric acid are well known plasticizers, triethyl citrate was introduced in PVC through a chemical bond. In this case, PVC was azide-functionalized to obtain an attached polymer. As expected, the migration of plasticizers was equal to zero. Mechanical properties of the material with modified PVC were compared to a material plasticized by DOP. The commonly used plasticizer provided significantly lower tensile strength and higher elongation at break than the modified one. Similar studies were carried out with the use of monooctyl phthalate, but the mechanical properties were not investigated [101].

In 2019, P. Jia et al. [62] proposed a PVC modification using bio-based plasticizers based on cardanol, castor oil and ESBO. A graft modification of PVC was conducted with thiosalicylic acid. Although there was no plasticizer migration from the plastics, the mechanical properties were still poor in comparison to PVC/DEHP. Later, in 2020, Najafi and Abdollahi [102] proposed to covalently attach to PVC chains four different compounds, namely tributyl citrate, propargyl ether tributyl citrate, oleic acid and poly(dimethylsiloxane) diglycidyl ether terminated. The T_g_ of all samples was lower than for neat PVC; however, the values were relatively high (62.6 to 41 °C) in comparison to blends with plasticizers that are not covalently bonded with PVC (in this study, such a comparison was not shown). The lowest T_g_ was exhibited by PVC with tributyl citrate, whereas the best performance was noted for PVC with poly(dimethylsiloxane) diglycidyl ether terminated. Elongation at break of modified PVC was higher than for neat PVC, and the highest value was obtained also when tributyl citrate was attached to PVC. The migration studies of PVC with attached tributyl citrate, as expected, confirmed no migration of plasticizer from the plastic. The limitation of this study is the lack of a comparison of mechanical properties using PVC blends with commonly used plasticizers or with compounds introduced in this study but without attaching them covalently to the PVC chain.

The triazole chemistry was applied to modify the PVC with triazole analogues of phthalate plasticizers [103,104,105]. A copper catalyzed or copper-free cycloaddition reaction between azide bearing PVC and alkyne groups containing branched and linear alkyl or polyether phthalate esters resulted in the formation of a triazole linkage. The nonmigratory internal plasticizer mimics were covalently attached to the PVC backbone as a pendant group. The effect of a variation of plasticizer structure, especially the ester moieties (different lengths of alkyl and polyether chains), on the internal plasticization was evaluated. The determined glass transition temperature varied from 96 °C (anti-plasticizing effect) to −42 °C (highly efficient plasticizer).

An interesting approach to the internal PVC plasticization through the block copolymerization of PVC with a more flexible polymer was proposed by Coelho et al. [106] Incorporation of a flexible, middle poly(n-butyl acrylate) block into PVC chains can be an alternative method to fabricating plasticized materials with similar properties to classically plasticized ones [107]. However, please note that this approach requires special conditions for polymerization, the so-called “living polymerization”.

The copolymerization approach to the nonmigratory internal plasticization was further developed in a recent research study of this group, where the PVC plasticization was achieved through the free radical copolymerization of vinyl chloride with 4,5-bis(2-ethylhexyl)-1-[6-prop-2-enoyloxy)hexyl]-1H-1,2,3-triazole-4,5-dicarboxylate (DEHT-HA) [108]. The DMA analysis of a PVC-co-P(DEHT-HA) copolymer with 75% content of vinyl chloride showed a single T_g_ (46.8 °C) that was lower than expected from the study of pure components. The acquired results implied that both segments were well miscible, and some synergistic effect was achieved. Varying the monomer ratio allowed us to tune the T_g_ of the resulting copolymer in a range between −27 and 78 °C [108].

## 5. Conclusions

The main problem in developing new plasticizers is to find a compound providing good mechanical properties to PVC, but with limited migration, extraction resistance and low volatility. There are a lot of studies describing novel plasticizers that exhibit excellent migration resistance at the expense of the deterioration of mechanical properties. Since phthalate plasticizers have become a subject of legal restrictions in many countries, developing a robust plasticizer emerges as an urgent need. Unfortunately, when newly developed plasticizers are compared to conventional ones (DEHP, DOP and DOTP, as shown in Table 1), in most cases the properties of the new ones are worse. Another important factor is a production cost of the novel compounds that sometimes is incomparably higher than for commonly used plasticizers. In the last few years, more attention has been paid to bio-based plasticizers derived from renewable resources that are thought to be more environmentally friendly and non-toxic compounds. Moreover, polymeric plasticizers (a great part of them are bio-based plasticizers) play an important role in searching for new compounds in areas with higher restrictions concerning migration resistance like childcare articles, food packaging or medicine.

Another trend to provide plasticized PVC material is to partially replace commonly used plasticizers with secondary ones. In this case, polymeric and low molecular weight plasticizers are frequently introduced. Interestingly, the majority of studies described the synergistic effects when a small amount of a secondary plasticizer is added, resulting in excellent migration resistance. The most advantageous effects are obtained when the mixture of plasticizers exhibits better migration resistance or elongation at break than for each one individually. Such an effect was obtained for PVC plasticized with GEHTMA-3 and SOHE, whereas incorporating epoxidized WCOEtHEs into a mixture with DOP in a PVC blend caused a higher elongation and a higher extraction in kerosene and migration resistance than in each one individually. Due to this notable improvement, these three plasticizers can be excellent secondary plasticizers.

The area of covalently bonded plasticizers seems to be still not fully explored, although they are very interesting due to “by definition” zero migration of such plasticizers. Only a few works on this subject have appeared in recent years and a lot can still be done to improve their plasticization efficiency.

Note: The commercial use of di(n-octyl) phthalate (DOP) is rare [15], so presumably in most cases where researchers are writing about DOP, in fact they are using DEHP, e.g., in [19,56,59]. In this review we use the same abbreviations as the authors’ original nomenclature. In 2006 Lindström and Hakkarainen proposed new plasticizers and compared them to both: DOP and DEHP. This study proved that these two compounds vary from each other and have slightly different physical properties, e.g., blends with DOP exhibit higher strain at break (562%) and lower stress at break (DOP—17 MPa) than comparable blends with DEHP (533% and 17.6 MPa, respectively) [21].

## Figures and Tables

**Figure 1 materials-14-00844-f001:**
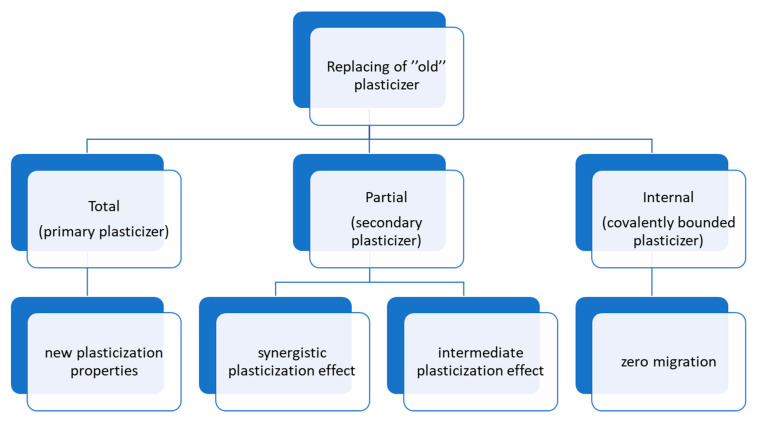
Strategies applied to replace the common “old” phthalate plasticizers.

**Figure 2 materials-14-00844-f002:**
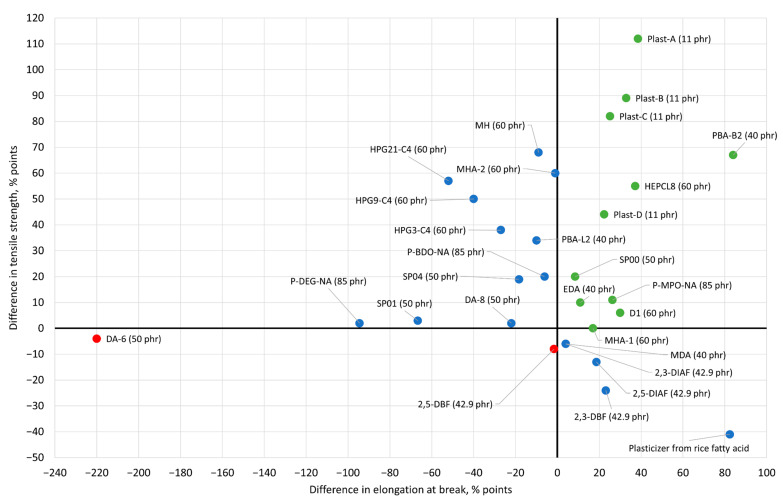
Correlation between elongation at break and tensile strength of the newly proposed plasticization compounds normalized to the analogues values obtained for PVC blends with the same amounts of a commercial reference plasticizer (as defined in Table 1) and expressed in the percentage points as a difference between the reference and the examined system. Green points indicate the relative increase in both the elongation at break and the tensile strength. Red points correspond to plasticizers that demonstrate lower values of both parameters, and blue points are reserved for plasticizers, which show an increase in only one parameter.

**Figure 3 materials-14-00844-f003:**
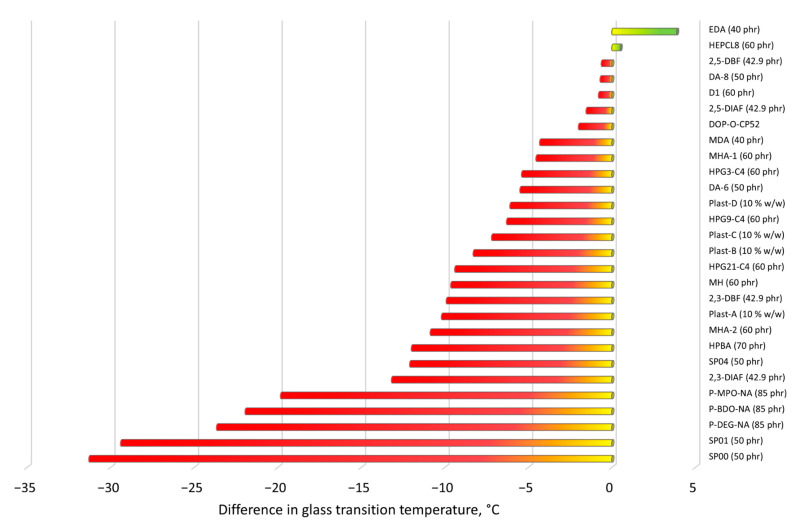
Difference in the glass transition temperature T_g_ between examined new plasticizers and PVC blends with the same amounts of a commercial reference plasticizer (T_g ref_—T_g exam._) (as defined in Table 1). The T_g_ of the reference blend was assumed to be “zero”. Green bars indicate plasticizers with a lower T_g_ than the reference system (only two cases). The majority of examined plasticizers exhibit a reasonable slight increase in T_g_; however, it should be kept in mind that they often benefit in other properties, especially in reduced migration or higher elongation at break, as shown in Table 1.

**Figure 4 materials-14-00844-f004:**
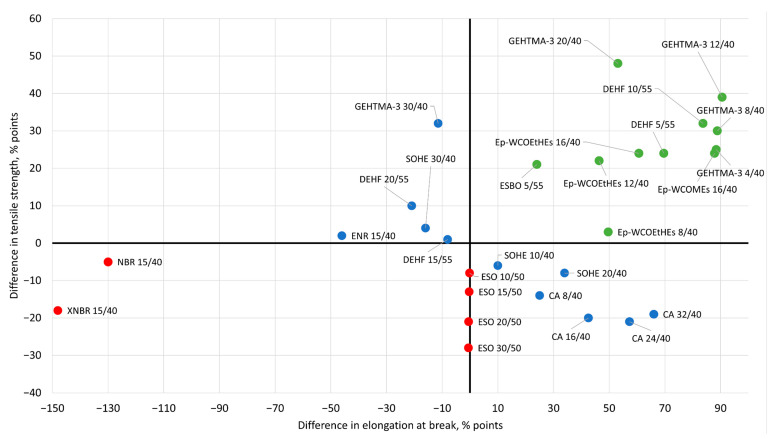
Correlation between the elongation at break and tensile strength of newly proposed mixed plasticization systems consisting of primary and secondary plasticizers normalized to the analogues values obtained for PVC blends with the same amounts of only primary plasticizers and expressed in the percentage points as a difference between the reference and the examined system. Green points indicate the relative increase in both the elongation at break and tensile strength. Red points correspond to the secondary plasticization systems that demonstrate lower values of both parameters, and blue points are reserved for plasticizers, which show an increase in only one parameter.

**Figure 5 materials-14-00844-f005:**
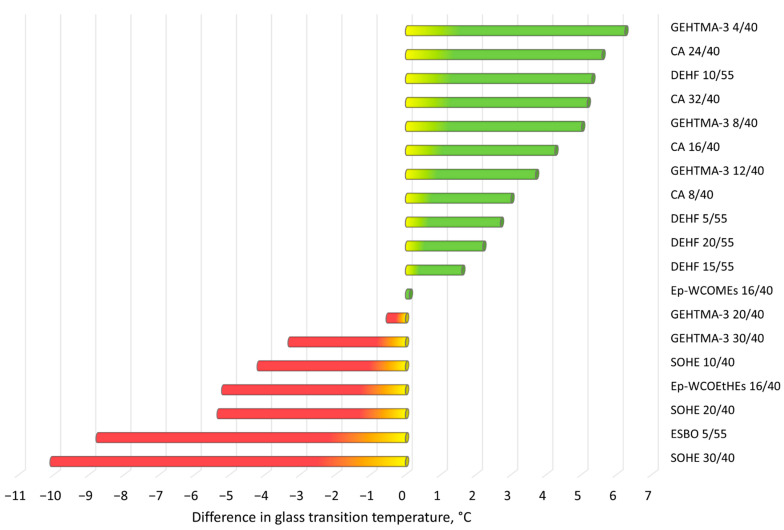
Difference in glass transition temperatures T_g_ between examined mixed plasticization systems consisting of primary and secondary plasticizers and reference PVC blends with the same amounts of only primary plasticizers (T_g ref._—T_g exam._). The T_g_ of the reference blend was assumed to be “zero”. Green bars indicate plasticizers with a lower T_g_ than reference system.

**Table 1 materials-14-00844-t001:** Comparison of properties of investigated primary plasticizers with common commercial plasticizers in terms of mechanical properties, glass transition temperature and migration of plasticizers.

Plasticizer Abbreviation and Content in the Mixture	Compared to Common Plasticizer (at the Same Content)	Tensile Strength Compared to Common Plasticizer (%)	Relative Elongation at Break (% points)	Glass Transition Temperature of Blend with Plasticizer (°C)		Migration Test Type and Conditions	Weight Loss of Blends with Plasticizer (%)		References
				Reference	Examined		Reference	Examined	
PBA-L2 (40 phr)	DEHP	134	10 ↓	cannot be estimate with enough accuracy		ageing in water, 37 °C, 10 weeks	no data	1.0	[21,48]
PBA-B2 (40 phr)		167	84 ↑					0.5	
HPBA (70 phr)	DOP	152	no data	11	23	exudation test, 60 °C, 30 days	3.75 ^b,^*	0.60 ^b,^*	[64]
D_1_ (60 phr)	DEHP	106	30.0 ↑	−15.5	−14.3	extraction in petroleum ether, 23 °C, 7 days	83.2 *	1 ^b,^*	[23]
HEPCL_8_ (60 phr)	DOP	155	37.19 ↑	−15.2	−15.7	extraction in petroleum ether, 23 °C, 7 days	88 ^b,^*	near 0 ^b,^*	[50]
HPG3-C4 (60 phr)	DEHP	138	27 ↓	−23.8	−18.4	leaching in n-hexane, 50 °C, 2 h	93.9 *	12.4 *	[49]
HPG9-C4 (60 phr)		150	40 ↓		−17.5			6.2 *	
HPG21-C4 (60 phr)		157	52 ↓		−14.4			2.0 *	
SP00 (50 phr)	DOTP	120	8.46 ↑	23.7	55	extraction in cyclohexane, room temperature, 24 h	19.2 *	0.3 *	[66]
SP01 (50 phr)		103	66.71 ↓		53.1			0.2 *	
SP04 (50 phr)		119	18.35 ↓		35.8			0.1 *	
DA-6 (50 phr)	DOTP	96 ^b^	220 ^b^↓	34.7	40.2	extraction in hexane, 23 °C, 24 h	31.6 ^b^	near 0 ^b^	[24]
DA-8 (50 phr)		102 ^b^	22 ^b^↓		35.4		31.6 ^b^	near 0 ^b^	
Plast-A (11 phr)	DEHP	212 ^a^	38.56 ↑	58^c^	68.2^c^	leaching in n-hexane, 50 °C, 7 days	12.2 ^d^	2.5 ^d^	[86]
Plast-B (11 phr)		189 ^a^	32.9 ^b^↑		66.3^c^			3.2 ^d^	
Plast-C (11 phr)		182 ^a^	25.2 ^b^↑		65.2^c^			7.0 ^d^	
Plast-D (11 phr)		144 ^a^	22.4 ^b^↑		64.1^c^			9.2 ^d^	
MDA (40 phr)	DOP	94	4 ↑	40.14	44.46	extraction in n-hexane, 24 h	6.8 ^b^	1.8 ^b^	[88]
EDA (40 phr)		110	11 ↑		36.26			2.9 ^b^	
Plasticizer from rice fatty acid	non plastified (pure) PVC	59 ^e^	82.33 ^e^	no data		no data			[89]
CDPac-ep1 (70 phr)	DEHP	233	0.1 [mm/mm]	no data		Ageing test 105 °C, 11 days	0.21 ^b^ [g]	0.24 ^b^ [g]	[82]
2,5-DBF (42.9 phr)	DOP	92	1.56 ↓	19.86	20.49	Extraction in n-hexane, 49.5 °C, 2 h	14.14	13.54	[91]
2,5-DIAF (42.9 phr)		87	18.62 ↑		21.40			13.10	
2,3-DBF (42.9 phr)		76	23.13 ↑		29.77			5.39	
2,3-DIAF (42.9 phr)		94	4.02 ↑		33.05			7.89	
MH (60 phr)	DOP	168 ^b^	9 ^b^↓	−22.05	−12.41	extraction in n-hexane, 25 °C, 24 h	19 ^b^	1 ^b^	[22]
MHA-1 (60 phr)		100 ^b^	17 ^b^↑		−17.51			1 ^b^	
MHA-2 (60 phr)		160 ^b^	1 ^b^↓		−11.18			1 ^b^	
P-MPO-NA (85 phr)	DEHP	111	26.34 ↑	−38.23	−18.40	extraction in n-hexane, 50 °C, 7 days	46 ^b,^*	13^b,^*	[26]
P-BDO-NA (85 phr)		120	6.13 ↓		−16.28			25 ^b,^*	
P-DEG-NA (85 phr)		102	94.48 ↓		−14.57			26 ^b,^*	
DOP-O-CP52 (32 phr)	DOP	No data	no data	23 ^b^	25 ^b^	extraction in n-heptane, room temperature, ca. 30 h	97 *	near 0 *	[96]
SNS/ESBO (1:1 wt%)	DTDP/TIMTM (2:3 *w/w*)	-	39 ^b^↑	no data		exudation, 136 °C, 168 h	near 0 ^b^	near 0 ^b^	[84]

^a^ in comparison with different contents of DEHP plasticizer—18 phr; ^b^ values estimated from the original graph; ^c^ plasticizer content in PVC blend: 25 phr; ^d^ plasticizer content in PVC blend: 18 phr; ^e^ because of a comparison only with pure PVC, data are not included in Figure 2; * degree (extend) of migration; ↑—improvement or ↓—deterioration.

**Table 2 materials-14-00844-t002:** Comparison of properties of the investigated mixed plasticization system that contains both primary and secondary plasticizers with the reference samples prepared only with a primary plasticizer in terms of mechanical properties, glass transition temperature and the migration of plasticizers.

Primary Plasticizer	Secondary Plasticizer	Total Amount of both Plasticizers (phr)	Amount of Secondary Plasticizer (phr)	Relative Improvement in Elongation at Break (% Points)	Glass Transition Temperature of Blend with Plasticizer (°C)		Migration Test Type and Conditions	Weight Loss of Blends with Plasticizer (%)		References
					Reference	Examined		Reference	Examined	
ECA	ESBO	55	5	24 *↑	27.4 **	18.6	no data			[25]
DEHP	NBR	40	15	130 ^a^↓	no data		leaching in petroleum ether, 30 °C, 72 h	23 ^a^	19 ^a^ (DEHP)	[97]
	XNBR		15	148 ^a^↓	no data			23 ^a^	20 ^a^ (DEHP)	
	ENR		15	46 ^a^↓	no data			23 ^a^	17 ^a^ (DEHP)	
DOTP	GEHTMA-3	40	4	88.47 ↑	43.90	37.66	extraction in petroleum ether, 24 h	13.2 ^a^	5.4 ^a^	[57]
			8	88.88 ↑	43.90	38.89		13.2 ^a^	3.6 ^a^	
			12	90.58 ↑	43.90	40.20		13.2 ^a^	2.4 ^a^	
			20	53.13 ↑	43.90	44.45		13.2 ^a^	1.8 ^a^	
			30	11.46 ↓	43.90	47.24		13.2 ^a^	0.8 ^a^	
DOP	SOHE	40	10	10 ^a^↑	41.46	45.68	extraction in petroleum ether, 23 °C, 24 h	11.8 ^a^	8.0 ^a^	[56]
			20	34 ^a^↑	41.46	46.82		11.8 ^a^	7.7 ^a^	
			30	16 ^a^↓	41.46	51.57		11.8 ^a^	7.0 ^a^	
DEHP	ESO	60	10	0.13 ^a^↓	no data		no data			[55]
			15	0.25 ^a^↓						
			20	0.45 ^a^↓						
			30	0.53 ^a^↓						
DOTP	CA	40	8	25.0 ↑	41.52	38.52	no data			[54]
			16	42.5 ↑	41.52	37.27				
			24	57.3 ↑	41.52	35.93				
			32	66.0 ↑	41.52	36.35				
DEHT	DEHF	55	5	69.62 ↑	23.5 ^a^	20.8 ^a^	leaching in cyclohexane, 23 °C, 48 h	21.6 ^a^	17.1 ^a^	[47]
			10	83.70 ↑	23.5 ^a^	18.2 ^a^		21.6 ^a^	19.8 ^a^	
			15	8.01 ↓	23.5 ^a^	21.9 ^a^		21.6 ^a^	15.5 ^a^	
			20	20.88 ↓	23.5 ^a^	21.3 ^a^		21.6 ^a^	12.5 ^a^	
DOP	Ep-WCOEtHEs	40	8	49.67 ↑	43.65	No data	extraction in kerosene, 23 °C, 24 h	0.4 ^a^	0.3 ^a^	[81]
			12	46.38 ↑	43.65	No data		0.4 ^a^	0.3 ^a^	
			16	60.69 ↑	43.65	48.88		0.4 ^a^	0.2 ^a^	
	Ep-WCOMEs		16	87.86 ↑	43.65	43.54		0.4 ^a^	0.5 ^a^	

^a^ value estimated from the graph; * in comparison to 50 phr of ECA; ** blend with 50 phr of ECA; ↑—improvement or ↓—deterioration.

## Data Availability

No new data were created or analyzed in this study. Data sharing is not applicable to this article.

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
