# Peer review of "Recent Attempts in the Design of Efficient PVC Plasticizers with Reduced Migration"

_materials, 2021, doi:10.3390/ma14040844_

Round 1

Reviewer 1 Report

The manuscript by Czogala et al. is on a very interesting and hot topic. The manuscript is written in a reader-friendly manner and the topics that are covered are very interesting. I have, however, some suggestions:

  • The manuscript should present a part in which the interactions of the plasticizers (external) with the PVC chains are illustrated. This will be important to allow the reader to better understand the different mentioned works. If possible, and if it is said in the original publication, the authors should indicate the type of interactions that are established between PVC and the plasticizer.
  • Section 2.1 is devoted to branched and hyperbranched polyesters, but the majority are also biobased. Thus, it does not make sense to name the section 2.2 as ‘Bio-based plasticizers’. I would advise the authors to change the title of section 2.2 to ‘Vegetable oils based plasticizers’.
  • Section 4 should be completed with the works of Coelho et al (Dep. Chemical Engineering, University of Coimbra) on the internal plasticization of PVC.

Author Response

Please find the response in the attached file.

Reviewer 2 Report

Monday the 18 of January 2021

Review on the manuscript "Recent attempts in the design of efficient PVC plasticizers with reduced migration" submitted to materials (IF=3) as a review.

The basic goal of the authors is to try and explain the development of new plasticizer for PVC.

The overview is fairly interesting, and the authors manage to extract the main features in the literature.
The manuscript is overall rather well the balance between the different section and subsections.

1. Introduction ---------------------------------------------------------------------------- 1
1.1. Plasticization mechanism ------------------------------------------------------ 2
1.2. Chemical classes of plasticizers ---------------------------------------------- 2
1.3. Migration of plasticizers ----------------------------------------------------- 2
1.4. General strategies to replace conventional plasticizers ----------------------- 3
Figure 1. Strategies applied to replace the common “old” phthalate plastizizers. --- 5
2. New primary plasticizers ---------------------------------------------------------------- 5
2.1. Branched and hyperbranched compounds ------------------------------------------ 5
2.1.1. Branched and hyperbranched compounds derived from adypic acid ------- 5
2.1.2 epsilon-Caprolactone based branched and hyperbranched compounds ------ 6
2.1.3. Other branched and hyperbranched plasticizers ----------------------- 7
2.2. Bio-based plasticizers -------------------------------------------------------- 8
2.3. Other primary plasticizers ------------------------------------------------- 10
Table 1. Comparison of properties------------------------------------------ 12
Figure 2. Elongation at break of newly proposed plasticization ------------ 16
Figure 3. Difference in glass transition temperature Tg ------------------- 17
3. New secondary plasticizers ------------------------------------------------------------- 18
Table 2. Comparison of properties ------------------------------------------------- 21
4. Internal plasticizers ------------------------------------------------------------------ 27
5. Conclusions ---------------------------------------------------------------------------- 27

So one has to agree with the structure of the manuscript. The references have also been choosen with care and a good balance between the "old" ones and the novelty.
The paper is rather clear to read and understand. I am pretty sure it will be read and quoted.
The quality of the graph is extremelly poor and they should be revised and presented in a modern manner (clear, good-looking, easy to understand) prior to publication.
Besides this problem that needs to be fixed, I would recommend the publication in the current form.

Author Response

(The authors gave the same response as above.)

Round 2

Reviewer 2 Report

Friday the 29 of January 2021 
Review on the manuscript "Recent attempts in the design of efficient PVC plasticizers with reduced migration" submitted as a revision to Materials.
The authors were very serious in changing their manuscript to account for the reviewers' comments. 
I can now recommend the publication in the current form. 
I wish this article will now be widely read to receive the numerous quotation it deserves.